# Capillary-Driven Microdevice Mixer Using Additive Manufacturing (SLA Technology)

Victor H. Cabrera-Moreta [1,2,*] and Jasmina Casals-Terré [1]

1    Laboratory of Microsystems & Nanotechnology, Mechanical Engineering Department, Polytechnic University of Catalonia (UPC), Colom Street 11, 08222 Barcelona, Spain; jasmina.casals@upc.edu
2    Mechanical Engineering Department, Universidad Politécnica Salesiana, Quito 170517, Ecuador
*    Correspondence: victor.cabrera.moreta@upc.edu or vcabrera@ups.edu.ec

**Abstract:** This study presents a novel microfluidic mixer designed, fabricated, and characterized using additive manufacturing technology — stereolithography (SLA) — and harnessing capillarity principles achieved through microstructure patterning. Micromixers are integral components in optimizing mixing and reaction processes within microfluidic systems. The proposed microdevice employs a tank mixing method capable of blending two fluids. With a channel length of up to 6 mm, the process time is remarkably swift at 3 s, and the compact device measures $35 \times 40 \times 5$ mm. The capillarity-driven working flow rates range from 1 µL/s to 37 µL/s, facilitated by channel dimensions varying between 400 µm and 850 µm. The total liquid volume within the device channels is 1652 mL (6176 µL including the supply tanks). The mix index, representing the homogeneity of the two fluids, is approximately 0.55 along the main channel. The manufacturing process, encompassing printing, isopropyl cleaning, and UV (ultraviolet) curing, is completed within 90 min. This microfluidic mixer showcases efficient mixing capabilities, rapid processing, and a compact design, marking it as a promising advancement in microfluidic technology. The new microfluidic mixer is a major step forward in microfluidic technology, providing a cost-effective and flexible solution for various uses. Its compatibility with SLA additive manufacturing allows for quick prototyping and design improvements, making it valuable for research and practical applications in chemistry, biology, and diagnostics. This study highlights the importance of combining advanced manufacturing techniques with basic fluid dynamics to create effective and easy-to-use microfluidic solutions.

**Keywords:** micromixer; additive manufacturing; mix index; SLA





## 1. Introduction

Effective mixing in a small channel and microfluidic devices are crucial in various scientific fields, like biomedical diagnostics, drug development, and various industries, due to their small flow channels that enhance the surface-to-volume ratios [1–5]. In chemical reactions, it is important to thoroughly mix the sample and reagent. If the mixing takes longer than the required reaction time, it hampers the performance of the reaction [5–8]. Therefore, in lab-on-chip or organ-on-chip devices, mixing is needed. The purpose of micromixing is to achieve a thorough, rapid, and efficient mix process in microscale device, despite the difficulties in the laminar behavior of the fluids in microdevices [1,2].

Active microdevices are systems incorporating sensors, actuators, or electronic components, designed to actively interact with their environment and perform specific functions. Devices with this technology include enhanced control and adaptability, with challenges such as increased complexity and potential issues in fabrication and integration [9–12]. On the other hand, passive micromixers have a low cost, a reduced power requirement, high portability, and more straightforward integration with other planar biomedical chips [13–15]. However, developing an accurate passive micromixer in a short manufacturing time would be challenging.

The efficient mixing of two fluids in a microchannel largely depends on the velocity of the fluids and the diffusion coefficient of the species. The longer the fluids remain in contact with each other, the greater the degree of mixing. However, it is also essential to consider that an excessively long microchannel can increase the pressure loss and design complexity [13–17]. Therefore, generally, passive micromixers using conventional soft lithography involve long channels or complex soft lithography processes with via holes. SLA additive manufacturing, also known as stereolithography, employs a laser to solidify a liquid resin, building layers to form 3D objects with high precision and detail. The possibility to use the three dimensions in the current SLA manufacturing methods allows an increase in the contact area between the mixing species by the design configurations of the microchannels [18,19]. However, recent examples show that good performance can be achieved, albeit using external pumping sources; see Table 1.

**Table 1.** The 3D-printed mixers driven by active methods.

| Active Methods | Application | Print Tech | Material | Channel Properties | Fluid Properties | Cover Material | Ref. |
|---|---|---|---|---|---|---|---|
| External pump | Red blood cell detection. | Polyjet | Clear acrylic | C = 200–300 L = 25 | FR = 15–20 MT = 3–9 | Polycarbonate membrane | 2021 [2] |
| Syringe pump | Monitoring organic reactions. | FDM | PLA | C = 700 L = 60 | FR = 25–100 MT = 18–71 | Closed | 2022 [1] |
| Dual syringe pump | Comparing microfluidic performance | FDM Polyjet SLA | Crystal clear Clear acrylic Clear resin | C = 300–1500 L = 27 | FR = 25–100 MT = 1–146 | Glass | 2021 [20] |
| Syringe pump | Microfluidics for nanomedicines. | FDM | PP | C = 400–600 L = 60 | FR = 1000 MT = 0.6–1.3 | Tape | 2021 [21] |
| Syringe pump | Microreactor: Y-shape. Flow control. | Polyjet | Clear acrylic | C = 660 L = 600 | FR = 200–2000 MT = 8–78 | Clear acrylic | 2019 [22] |
| Syringe pump | Performance of 3D-printed mixers. | FDM Polyjet SLA | PLA White resin Clear resin | C = 600 L = 80–150 | FR = 10–400 MT = 4–173 | Tape | 2019 [18] |

C—Channel Size (µm), L—Length (mm), FR—Flow Rate (µL/min), MT—Mixing Time (s).

According to the literature review in the chart above and complementary information, researchers propose different mixing methods for 3D-printed microdevices, including active methods that require long-length channels (25–600 µm), a wide range of mixing times (0.6–146 s), or high flow rates (10 to 2000 µL/min) [18,23–31]. All the devices listed in the table above require an active method for their operation. Since they rely on an external component, the required speeds and pressures are high. The minimum channel length for proper fluid mixing execution is 25 mm. The novel propose of this study is to achieve results obtained with active devices using passive methods such as capillarity.

The low Reynolds number in microchannels results in limited turbulent mixing, making species' mixing primarily dependent on slow diffusion. This poses a challenge for efficient mixing within short channels, crucial for maintaining the miniature nature of microfluidic devices. Developing effective mixing schemes becomes essential not only for thorough species mixing but also for increased throughput in microfluidic systems and to achieve the concept of micro-total-analysis systems and lab-on-a-chip devices [1,2]. As a result, enhancing the fabrication method to achieve complex and novel geometries is a crucial aspect of the current research. This improvement would allow for the management of laminar fluids in small devices and an efficient mixing process.

Traditional methods, such as soft lithography and micro-milling, encounter challenges in optimizing the fabrication process, reducing the time, and achieving complex geometries, particularly in 3D [10,32,33]. The current manufacturing methods for microfluidic models require a considerable number of steps and specialized equipment to



develop new prototypes [9]. CNC milling requires high-precision equipment to manufacture microdevices, and the precision or minimum feature size cannot be reduced below 100 microns [9]. Both processes are time-consuming. There is a need for a new manufacturing method to achieve a faster microfluidic manufacturing process.

SLA technology is proposed, enabling the precise fabrication of microchannels and chambers, not only in-plane but also out-of-plane [34–36]. This capability enhances the incorporation of capillary-driven mixing mechanisms within the device, optimizing the use of smaller areas. The manufacturing method for the mixer's design harnesses the spontaneous capillary flow of fluids along microscale channels, promoting rapid and efficient mixing without the need for an external system, as seen in conventional prototypes for point-of-care applications [37–39].

Recent studies have introduced 3D printing technology as a promising manufacturing tool for microdevices [40]. During the last few years, 3D printing, or additive manufacturing, has been introduced since it is a fast, simple, and low-cost technology to fabricate devices. There are current research projects that use 3D technology for microfluidic device fabrication in the areas of healthcare, chemistry, and engineering due to its simplicity and low cost [9,10,40]. The traditional manufacturing methods also limit the variety of materials used in the manufacturing process (glass, polymeric materials, PDMS, PMMA) [1]. Moreover, 3D printing increases the options for printing materials to provide the researcher with more options for experimentation [11]. However, the 3D printing process for microchannels has not been comprehensively studied. Consequently, there is an opportunity to research the improvement of 3D printing as a manufacturing process for micro-platforms.

It is necessary to devise a design mixer that improves the efficiency and reduces the time required to assemble a complete mixing platform. It is necessary to overcome the current manufacturing barriers of 2D micromixers using 3D approaches to achieve enhanced efficiency. The results of the study demonstrate the cost-effectiveness of rapid prototyping applied to micromixers, especially to those that are capillary-driven.

This research aims to analyze information from previous micromixer devices and establish a study limit. Ultimately, the goal is to propose a passive device suitable for use as a mobile lab-on-chip, eliminating the need for an external power source. This objective can be achieved through an alternative manufacturing process capable of creating complex geometries to exploit the laminar flow behavior common in microdevices. SLA technology is suggested as an accurate method of obtaining the planned results. Maintaining a compact size is crucial to ensure portability and ease of manipulation for the device.

## 2. Materials and Methods

### 2.1. Device Printing Process

The traditional and recommended SLA 3D printing process includes four main steps: digital design, printing, washing, and curing [12]. However, microchannels under 400 μm show material stagnation. As a result, an improvement to the process is added.

The process includes an air cleaning step between the stages of cleaning and curing. This stage, with alcohol and air, is repeated before the curing process, until the channel is clear.

Design. This stage involves developing the 3D model of the mixer using parametric software (SOLIDWORKS 2023). The 3D model is then exported as a stereolithography file (STL), which will be uploaded to the printer to generate the model.

Printing. For this stage, a FORM 3+ printer was used (manufacturer: Formlabs Inc., Massachusetts, USA). SLA technology was applied to manufacture the models. The resolution used on the print devices was 50 μm (printing resolutions available: 25, 50, and 100 μm [41]). The maximum build dimensions of the printer are $145 \times 145 \text{ mm}^2$ in width and 185 mm in height [42]. The CLEAR RESIN by FORMLABS was used to print the devices. Figure 1 illustrates the outcomes obtained with various types of resins: gray, black,

and clear. While the behavior across different materials exhibits similarities. The CLEAR material significantly enhanced the quality of images capturing the mixing process.

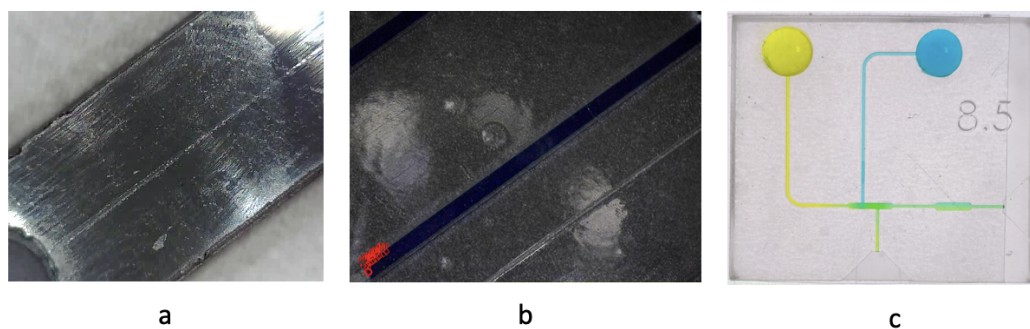

**Figure 1.** Different resins for printing: (**a**) gray resin, (**b**) black resin, (**c**) clear resin.

Cleaning. After printing, a mandatory cleaning process with isopropyl alcohol is required. The uncured resin (in a liquid state) can clog the channels. An automated washer (Form Wash (2nd Generation)) was utilized for this stage. The alcohol cleaning process was combined with air cleaning. The wash cycle takes 5 min in the washer; then, an air cleaning process with 4 psi pressure fluid is necessary. The device will repeat this cycle three times in the washer. After the final air cleaning step, the device will be cured.

Curing. A FORMCURE (curing chamber) was employed for this process. The curing chamber exposed the device to ultraviolet rays. The model was cured at 60 degrees Celsius for 12 min.

### 2.2. Device Setup and Data Collection

Figure 2 shows the process of device preparation, experiment execution, data acquisition, and analysis.

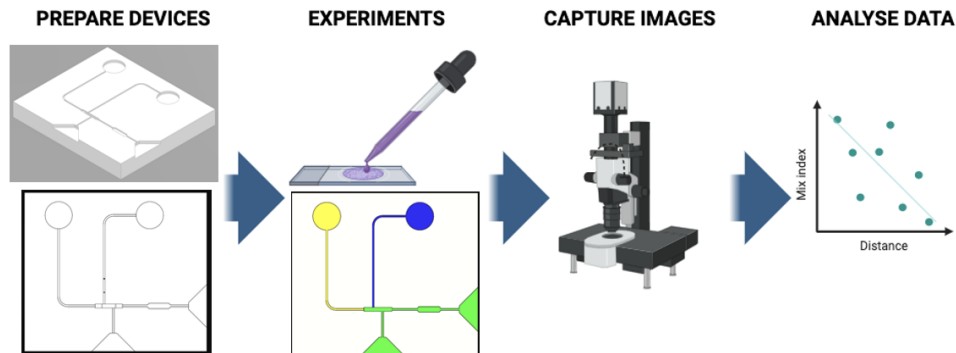

**Figure 2.** Data analysis process.

### 2.2.1. Device Preparation

Surface Treatment: To promote capillary flow, a hydrophilic adhesive, specifically designed for microfluidic applications and meeting the required specifications, is uniformly applied to the device's surfaces. This adhesive facilitates the optimal wetting of the channels and enhances the fluid flow. Specification: 3M 9984 Diagnostic Microfluidic Surfactant Free Hydrophilic Film.

Cleaning and Drying: The prepared device is meticulously cleaned with high-purity distilled water (grade 0) to remove any residual contaminants or particles. Subsequently, it is dried using a controlled stream of compressed air at 5 psi. This cleaning and drying process is performed before each new experiment to ensure consistent and reliable results.

### 2.2.2. Experimental Setup

Platform Leveling: Prior to each experiment, the microdevice is positioned on a precision-leveled platform to minimize potential disturbances caused by any tilting or unevenness.

Number of tests: Five devices were fabricated, which were reused after removing the adhesive from the top, cleaning with distilled water, and drying with compressed air (5 bar). The experiment was repeated three times on each device, resulting in a total of 15 tests.

Sample Preparation: A precisely measured 20 µL volume of high-purity distilled water (grade 0) is loaded into a calibrated pipette. This distilled water is pre-prepared with a biocompatible vegetable dye to visualize the fluid flow. Figure 3 shows the main parts of the manufactured device.

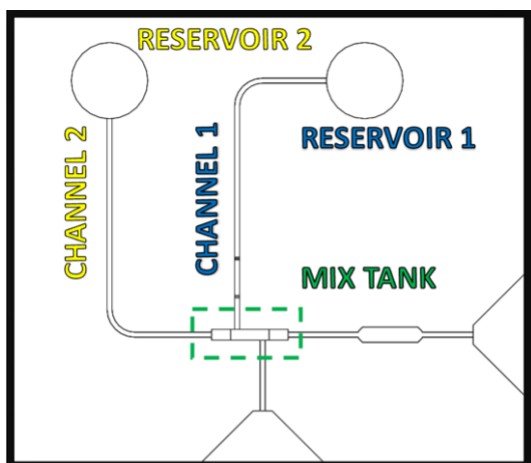

**Figure 3.** Data analysis process.

Color Placement and Sequential test: The experiment begins by introducing blue-colored distilled water into the designated reservoir 1 (Figure 5a). The capillarity-driven flow spontaneously initiates, propelling the blue fluid into the microchannel. Once the microchannel is sufficiently filled with the blue color, it stops at the end of the channel due to a trigger valve (TV). Figure 4 illustrates the detailed view of the TV (Trigger Valve) component within the device. Subsequently, yellow-colored distilled water is introduced into reservoir 2 (Figure 5b). At the end of the channel, it triggers the blue liquid to cross through the mix tank, initiating the sequential mixing process. This allows for the observation and analysis of the mixing efficiency as the two colored fluids merge and progress through the microchannel network.

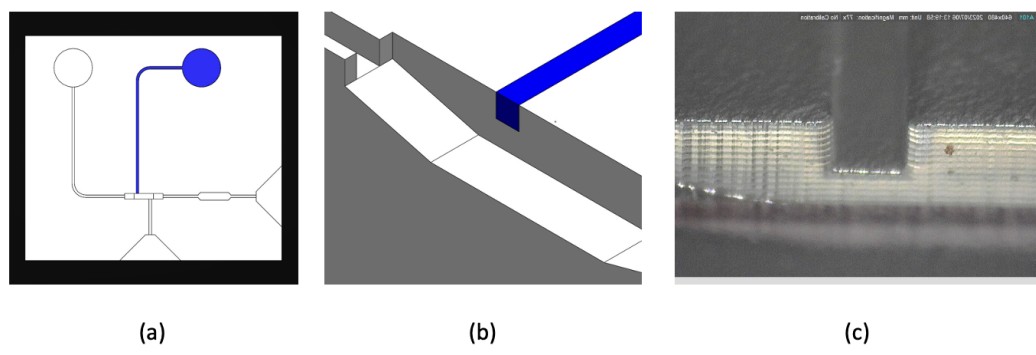

(a)　　　　　　　　　　(b)　　　　　　　　　　(c)

**Figure 4.** Trigger valve (TV). (**a**) Stop valve at end of channel 1, (**b**) 3D view of TV, (**c**) TV microscope image.

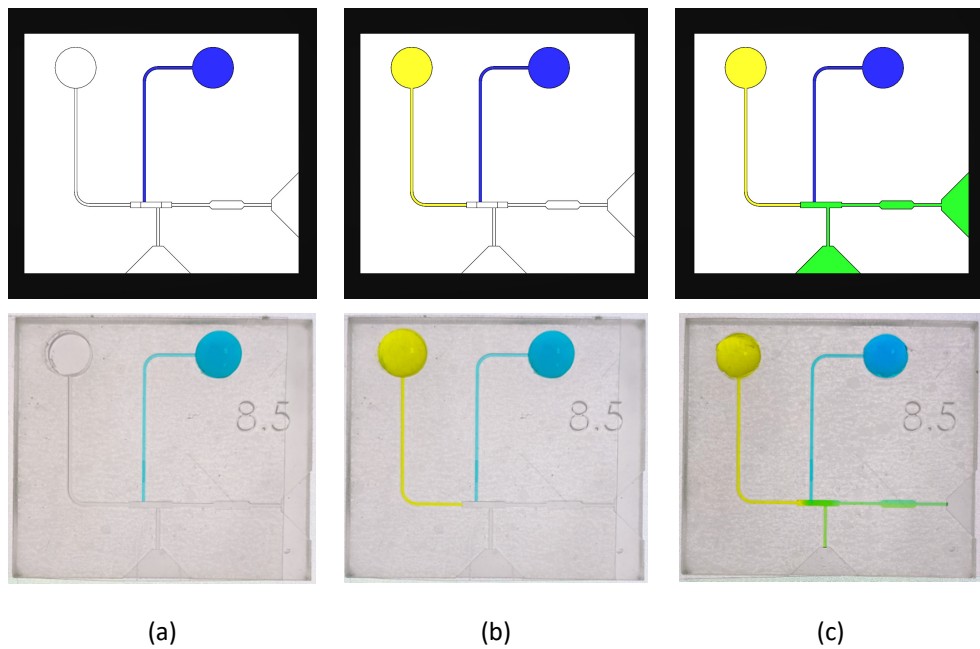

**Figure 5.** Referential image of mixing process. (**a**) Blue water load (channel 1), (**b**) yellow water load (channel 2), (**c**) mix process inside chamber (mix tank).

According to the literature review, a trigger valve (TV) effectively operates when the gap between the channel and the bottom of the main channel maintains a minimum difference of 300 µm [43]. In this configuration, the liquid in the small channel halts, awaiting activation, and is triggered once the main channel is filled with another liquid. In our current device, this gap measures 400 µm. The small channel is filled with blue distilled colored water, while the main channel contains yellow distilled colored water.

### 2.2.3. Video and Image Processing

Image and Video Capture: High-resolution images and videos were captured using a USB digital microscope (Dino-Lite Digital Microscope AF4515ZTL). To ensure the highest image quality, a smart cellphone was positioned underneath the device with a white light screen, optimizing the illumination for image and video capture. The prepared device was placed over the white screen light. The microscope was focused and positioned on top, as shown in the third stage in Figure 3. The experiment was then recorded with a sequence of images at 0.1 s of exposure. The video was recorded and saved in the repository for further analysis.

Software Utilization: The DINO CAPTURE 2.0 software was employed to save the image and video files, ensuring standardized and organized data storage. Additionally, it facilitated the real-time observation of the microfluidic processes. The experiment was recorded and digitized through images.

Contact Angle Measurement: The contact angles at the fluid interfaces were quantified using images obtained from the USB microscope. Multiple measurements (n > 3) were performed for statistical reliability. The analysis was carried out using the DINO CAPTURE 2.0 software, allowing for the accurate determination of the contact angles.

Channel size: To determine the real size of the channel, a perfilometer (Bruker brand, model Dektak XT) was used. It was validated by image correlation with the Dino-Lite Digital Microscope (AnMo Electronics Corporation, Taipei, Taiwan).

### 2.2.4. Data Analysis

Velocity Measurement: Velocity data were extracted from the videos using the free software TRACKER, a powerful tool for video analysis and modeling. This facilitated

the calculation of the fluid velocities within the microchannels. The videos and images obtained through the digital devices were loaded into the program, as shown in Figure 6b. With the images, the initial position of each fluid was established. Through detection algorithms, the software determined the position of the fluid in each sequence of images. The obtained data included the position in the × and y coordinates, as well as the execution time and the relative velocity of the fluid compared to the previously captured point.

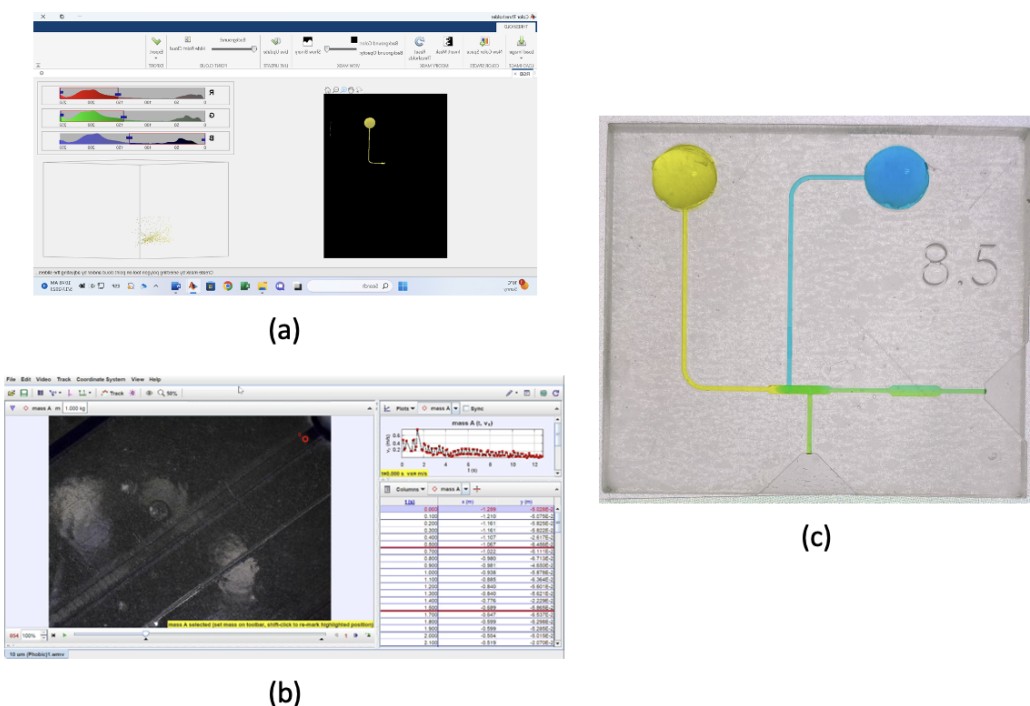

(a)

(b)

(c)

**Figure 6.** (**a**) Threshold color image tool using MATLAB, (**b**) TRACKER reference image, and (**c**) IMAGE mixing index analysis.

Graphical Representation: MATLAB R2023a was employed for data processing and the creation of graphical representations. Graphs and figures were generated to visualize the experimental results, aiding in data interpretation and comparison. The data tables were loaded into the program, and, through plotting codes, they were represented graphically to present them in the results of this work.

Color Analysis: Color analysis was conducted to evaluate the fluid mixing efficiency. The threshold color image tool within the MATLAB Toolstrip was used to precisely determine the color changes within the microchannel network. The final images captured by the USB microscope were loaded into the tool and filtered by color range. This analysis, as shown in Figure 6c, allowed the verification that the mixing process occurred adequately.

Mixing Analysis: ImageJ, a freely available software program, was used for the analysis of the mixing patterns within the microdevice. Images were imported into ImageJ, and the data were subsequently exported to MATLAB for further comparison with the simulation data. The software allowed us to obtain data from the images by comparing the color intensity in relation to the position of the fluid. With the obtained data, we were able to compare the tonality of each stage, both experimentally and through simulations.

By following this refined methodology, we ensured precise device preparation, optimal surface treatment, a controlled experimental setup, and precise data acquisition and analysis in our capillarity-driven microdevice mixer study. All of these are essential for accurate and replicable results in capillarity-driven microdevice mixing studies. The combination of advanced imaging techniques and powerful software tools enabled the accurate quantification of the contact angles, fluid velocities, color changes, and mixing patterns,

providing a thorough understanding of the device's performance and its comparison with the simulation data.

### 2.3. Simulation

To ensure the reliability and accuracy of our experimental findings, a computational fluid dynamics (CFD) simulation was conducted using the ANSYS software. Initially, a parametric model of the microdevice was meticulously crafted using SOLIDWORKS. This model was then seamlessly transferred to ANSYS using an IGES (Initial Graphics Exchange Specification) file. The CFD simulation was executed iteratively, with each iteration refining the model until the results achieved convergence. This iterative process allowed us to systematically fine-tune the simulation parameters and boundary conditions to closely match the real-world behavior observed in our experiments. By harmonizing the experimental data with the simulation outcomes, we were able to validate and enhance the comprehensiveness of our findings, contributing to a more robust and well-rounded analysis of the microdevice's performance. The parameters used in the simulation are summarized in Table 2.

**Table 2.** ANSYS parameters.

| Setup | Parameter |
| --- | --- |
| Multiphase Model | Volume of Fluid (explicit)<br>Number of Eulerian Phases: 2<br>Surface Tension Coefficient: 0.072 [N/m] |
| Viscous Model | k-epsilon (2 eqn) (Standard)<br>Standard Wall Function [N/m] |
| Velocity Inlet | Velocity magnitude of liquid 1: 0.004 [m/s]<br>Velocity magnitude of liquid 2: 0.001 [m/s] |
| Run Calculation | Number of Time Steps: 40<br>Time Step Size: 0.1 [s]<br>Max Iteration: 20 |

The software simulation method employed the following transport equations for the viscous model, utilizing the standard k-epsilon model (a two-equation model) along with the Standard Wall Function:

$$\frac{\delta}{\delta t}(\rho \kappa) + \frac{\delta}{\delta x_i}(\rho \kappa \mu_i) = \frac{\delta}{\delta x_j}\left[\left(\mu + \frac{\mu_t}{\sigma_\kappa}\right)\frac{\delta \kappa}{\delta x_j}\right] + G_\kappa + G_b - \rho \varepsilon - Y_M + S_\kappa \tag{1}$$

$$\frac{\delta}{\delta t}(\rho \epsilon) + \frac{\delta}{\delta x_i}(\rho \epsilon \mu_i) = \frac{\delta}{\delta x_j}\left[\left(\mu + \frac{\mu_t}{\sigma_\epsilon}\right)\frac{\delta \epsilon}{\delta x_j}\right] + C_{1\epsilon}\frac{\epsilon}{\kappa}(G_\kappa + C_{3\epsilon G_b}) - C_{2\epsilon}\rho\frac{\epsilon^2}{\kappa} + S_\epsilon \tag{2}$$

$\kappa$—turbulence kinetic energy;
$\epsilon$— rate of dissipation;
$G_\kappa$—turbulence kinetic energy due to mean velocity gradients;
$G_b$—turbulence kinetic energy due to buoyancy;
$Y_M$—fluctuating dilatation in compressible turbulence to overall dissipation rate;
$C_{1\epsilon}$ $C_{2\epsilon}$ $C_{3\epsilon}$—constants;
$\sigma_\kappa$ and $\sigma_\epsilon$—turbulent Prandtl numbers for $\kappa$ and $\epsilon$, respectively;
$S_\kappa$ and $S_\epsilon$—user-defined source terms.

## 3. Results

### 3.1. Printed Device

Following a series of iterative printing tests and refinements, we proudly present the finalized design of the proposed device, as depicted in the figure below. This culmination of multiple printing experiments represents our commitment to achieving the optimal

configuration and functionality of the device, ensuring that it meets the highest standards of performance and precision.

The gap between the main channels and mixing tank is 500 µm (Figure 7, Detail C). This gap is important because it allows us to stop the fluid to ensure that the mixing is sufficient. The picture above gives the most important information and measurements of the design.

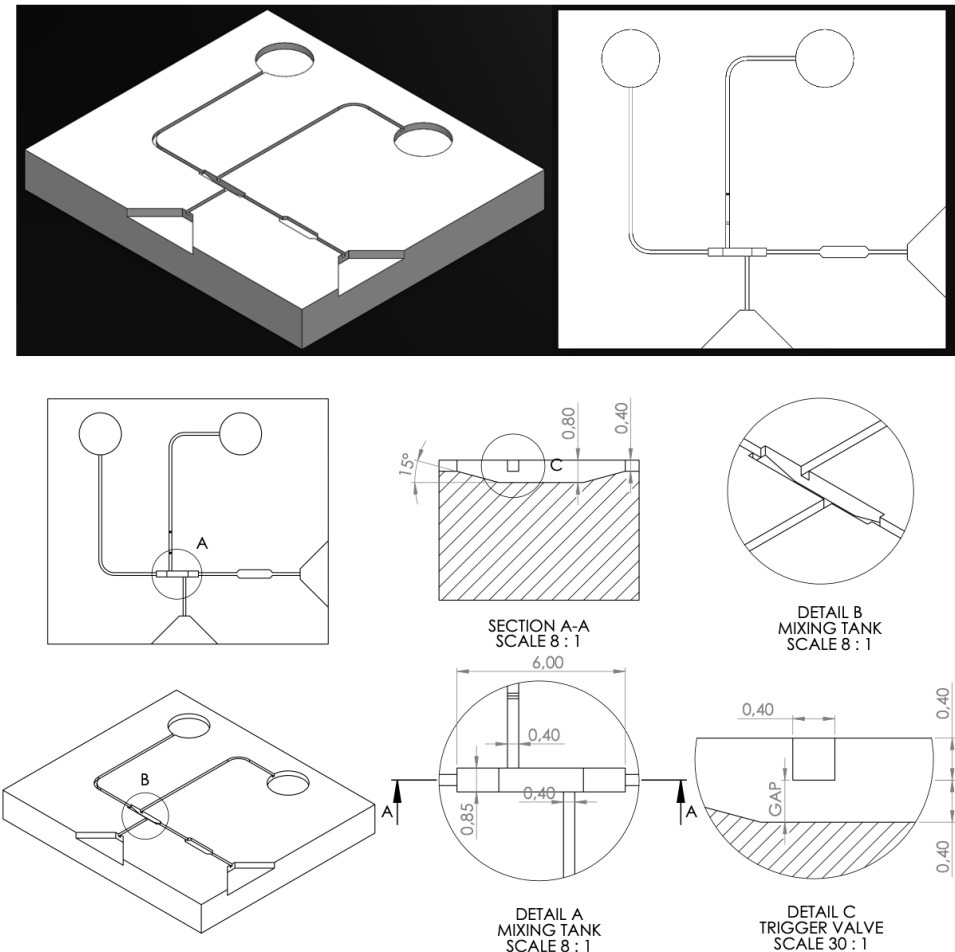

**Figure 7.** 3D design device with CAD. Detail A: Mix Tank top view. Detail B: Mix Tank 3D View. Detail C: Trigger Valve location.

Figure 8 displays SEM microscope captures of different areas within the mixing device. The measurement data of the acquired channels are presented in Table 3.

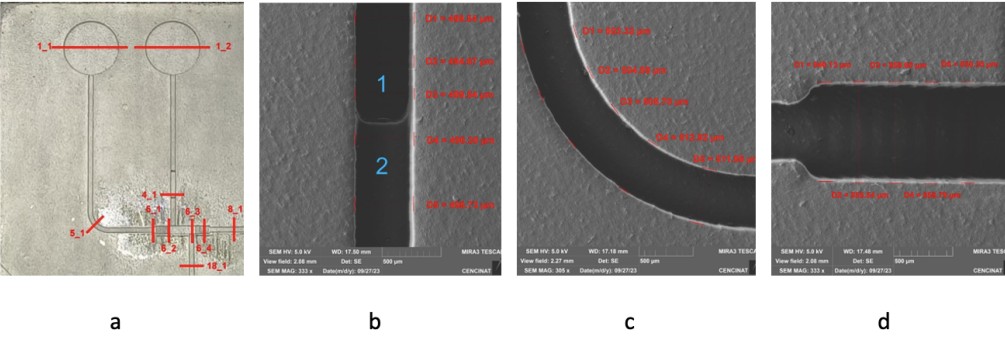

**Figure 8.** The 3D SEM images: (**a**) device zones, (**b**) zone 4, (**c**) zone 5, (**d**) zone 6.

**Table 3.** Measurements of the channels.

| Zone | Average (μm) | Deviation (μm) |
|------|--------------|----------------|
| 4 | 497.76 | 2.18 |
| 5 | 507.72 | 4.29 |
| 6 | 862.91 | 12.82 |

The main characteristics of the device are as follows.

- Scan time: 400 (s) for each zone.
- Length scan: 20,000 (μm) for zone 1 and 5000 (μm) for zones 2 and 3.
- Scan resolution: 0.0166667 (μm) for zone 1 and 0.0462954 μm for zones 2 and 3.
- Scan type: standard.
- Needle force: 3 (mg).
- Needle range scan: 1 (mm).
- Needle type: radius 25 (μm).
- Correction: quadratic and removal of curvature.

### 3.2. Velocity of Fluid during Experiment

Our research into the fluid velocities within the mixing device yielded insightful motion curves. These curves vividly demonstrate the impact of positioning the pause valve at the terminus of the yellow liquid channel. Specifically, this adjustment results in a notable reduction in fluid velocity, transitioning from an initial velocity of 13.45 mm/s to a diminished speed of 3.033 mm/s (Figure 9a). This observation highlights the effectiveness of the pause valve in modulating and controlling the fluid flow within the microdevice, an essential aspect in achieving precise and controlled mixing dynamics (Figure 9c).

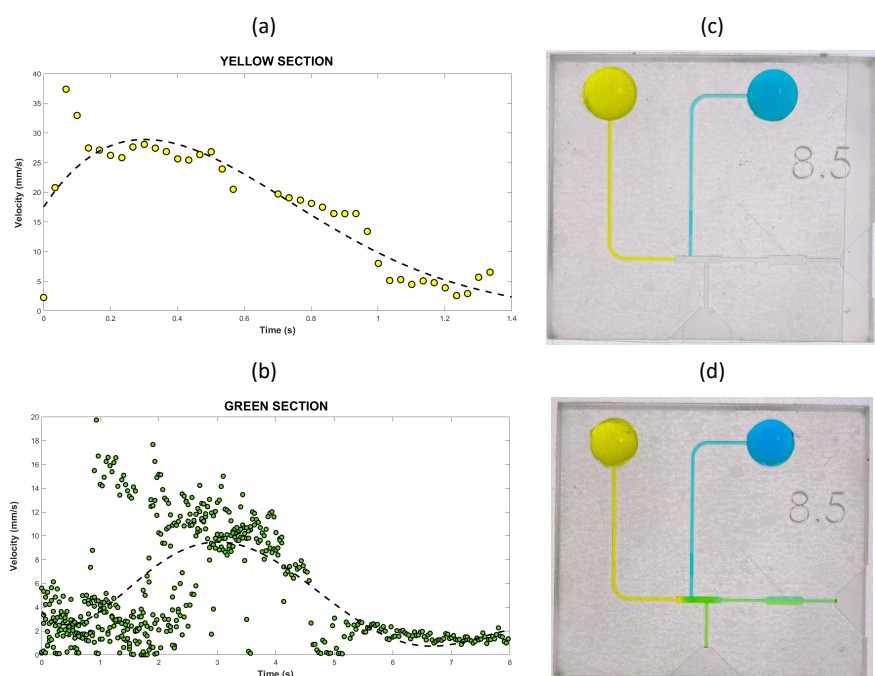

**Figure 9.** Fluid velocity during mixing process. (**a**) Velocity curve of the yellow liquid, (**b**) Velocity curve of the mixed liquid within the tank, (**c**) Image of the yellow and blue fluids at rest, (**d**) Image of the final result after mixing.

Once the fluids have undergone mixing, the resulting, green-colored solution exhibits distinct velocity patterns within the microdevice (Figure 9d). As it enters the mixing tank, it advances at a controlled rate of 3.96 mm/s, reflecting the balanced flow dynamics achieved during the mixing process (Figure 9b).

Continuing through the feeding channel, the fluid accelerates significantly, surging forward at a brisk pace of 22.97 mm/s. As the channel section narrows, the velocity further increases, reaching a peak of 15.09 mm/s, emphasizing the influence of the channel geometry on the fluid behavior. Ultimately, the fluid comes to a precise halt shortly before reaching the second mixing tank. These observed velocity variations provide valuable insights into the controlled fluid flow within the microdevice, contributing to a deeper understanding of its operational dynamics and efficiency in various applications.

### 3.3. Validation of Results

### 3.3.1. Mix Analysis

With the MATLAB software, including its dedicated color analysis tool, we were able to discern and differentiate the distinct stages of the mixing process. This analytical approach allowed us to visualize the mixing dynamics at each phase, and the visual evidence unequivocally indicates the remarkable efficiency and effectiveness of the mixing process achieved. Figure 10 demonstrates the image of the color analysis tool.

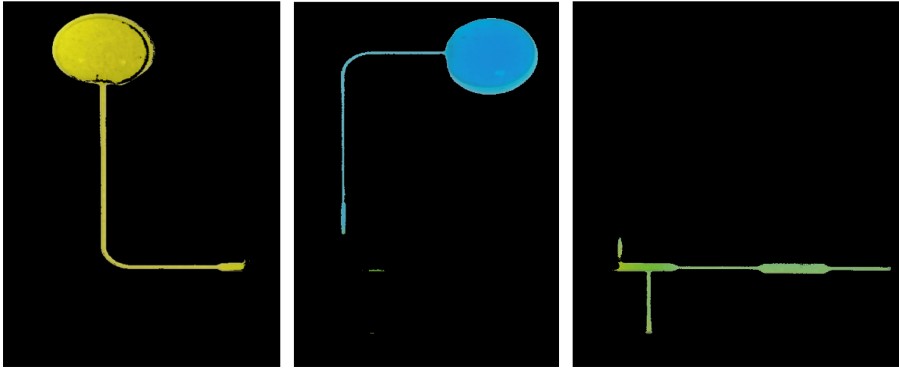

**Figure 10.** MATLAB analysis with color filter tool.

### 3.3.2. Simulation

To confirm and validate our experimental results, we conducted a rigorous simulation using the ANSYS platform, specifically employing the computational fluid dynamics (CFD) module. The parameters essential for running this simulation are meticulously outlined and summarized in the table provided below, offering a comprehensive view of the key factors and settings utilized in our analysis.

Figure 11 shows the mesh convergence for different arrangements. It can be concluded that values below 125 µm were suitable for an accurate simulation result. Additionally, the image shows the element quality obtained in the model.

Figure 12 displays the simulation results, which serve as a visual confirmation of the validated mixing process achieved through our software-driven analysis. These results provide a tangible and illustrative representation of the intricacies of the mixing phenomenon, reinforcing the reliability and accuracy of our computational assessments. The 850 µm width and 800 µm deep tank size ensure the most efficient mixing process. According to the results, it achieves a mix index of 055 considering the fluid at the main channel.

Figure 13 provides a visual representation of the results obtained from the mixing process within the tank of the device. This image encapsulates the culmination of the mixing operation, showcasing the intricate interplay of the fluids within the tank. It offers insights into the dynamic behavior of the substances being mixed, highlighting the effectiveness and efficiency of the mixing mechanism employed. By observing the distribution and interaction of the fluids within the mix tank, valuable information can be gleaned regarding the homogeneity, consistency, and thoroughness of the mixing process. Figures 13 and 14 clearly show the effectiveness of the simulation process compared with the experimental test.

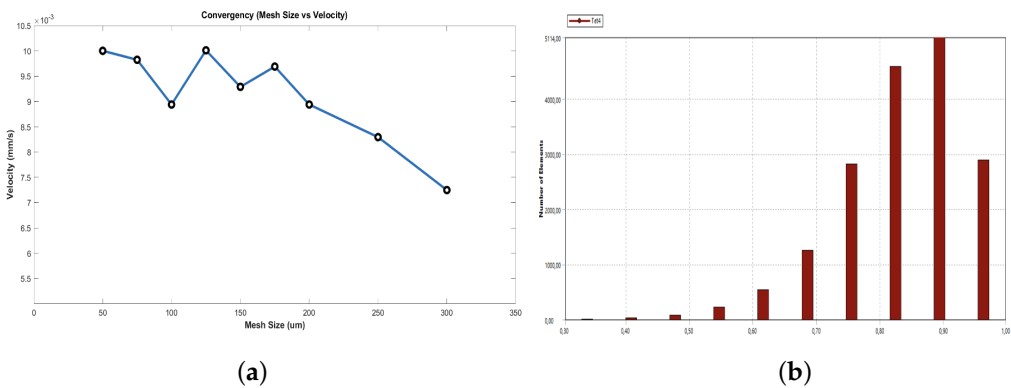

**Figure 11.** (**a**) Mesh size and convergence, (**b**) element quality.

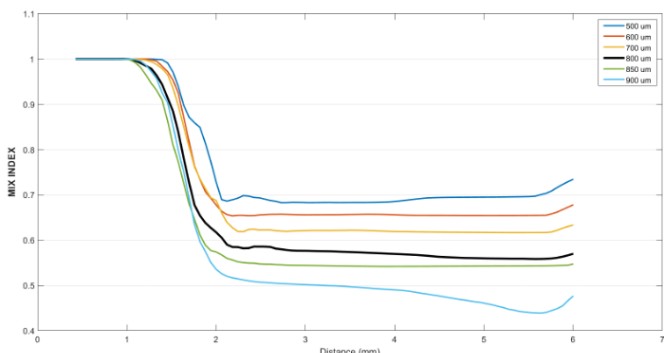

**Figure 12.** Mix index according to channel size variation.

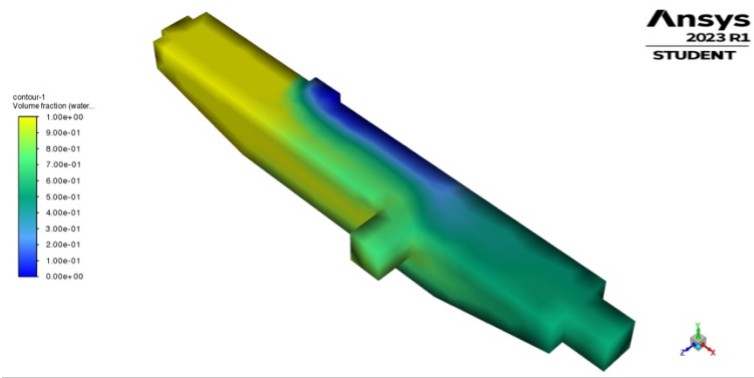

**Figure 13.** ANSYS image simulation result.

### 3.3.3. Comparison of Results

We examined the results of our computer simulation and compared them to a series of pictures from our actual experiments. When we evaluated them side by side, it was easy to see that the computer simulation matched the real experiment's results, confirming that our computer model can accurately predict the outcome.

Figure 14 displays five images of the mixing process in the simulation and experiments. The image focuses on the mixing chamber, as it is the location where the process is most relevant. The images were captured using a digital USB microscope (Dino-Lite Digital Microscope AF4515ZTL (AnMo Electronics Corporation, Taipei, Taiwan)). In the first stage, the blue fluid remains contained at the stop valve while the yellow fluid is introduced into the mixing tank. Progressing to the second stage, the yellow fluid reaches the stop valve, prompting its opening. Subsequently, the initiation of the mixing process is depicted as the blue fluid is released, causing both fluids to advance uniformly within the chamber. Moving to the fourth stage, an outlet duct is revealed, facilitating the controlled release of

the yellow fluid and ensuring thorough mixing within the chamber's final section. Finally, the mixing tank attains full capacity, indicating successful completion of the mixing process.

In the experimental graphs and simulation charts, we can observe the similarity in the process. Figure 15 illustrates the details of this process resemblance. Through this analysis, we can validate that simulation, in this case, is a valid tool for the design of future devices that operate based on similar principles.

### 3.3.4. Mix Index Analysis

We used both numerical results and pictures to compare the results that we obtained. In the graphs below, we use values from the image analysis to demonstrate the mixing process. We also compared these values with data from our computer simulation to ensure their matching. Figure 15 displays one random test datum from each device. Three samples from each of the five devices were used. The graph illustrates the repeatability of the experiment.

In this way, we can confirm that our simulation accurately represents the mixing process.

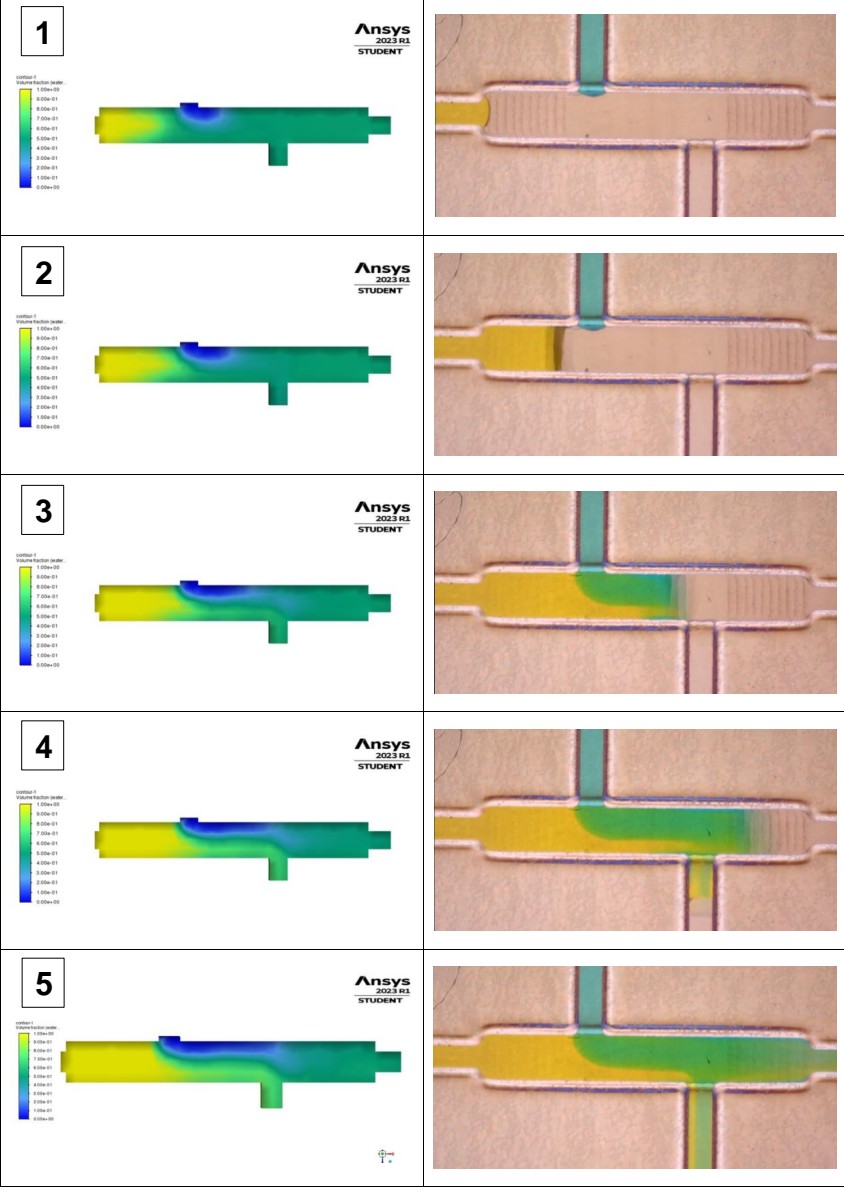

**Figure 14.** Comparative graph of ANSYS simulation and photo of the mixing process.

In summary, our study showcases the potential of SLA technology for the fabrication of microchannels and highlights the efficacy of capillary-driven mixing, validated through simulations. The diverse applications of our findings in various industries underscore the practicality of manufacturing mixing systems using SLA additive manufacturing. Moreover, the ability to achieve rapid mixing in short channels represents a compelling advancement in microfluidic technology, promising enhanced performance and efficiency for a wide range of applications.

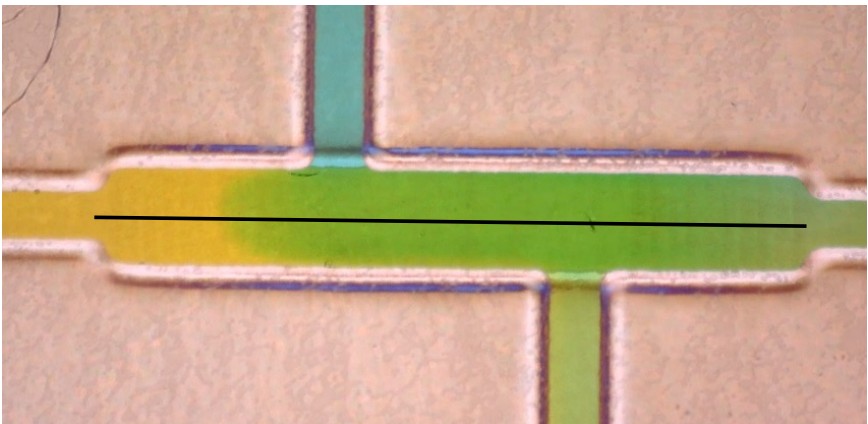

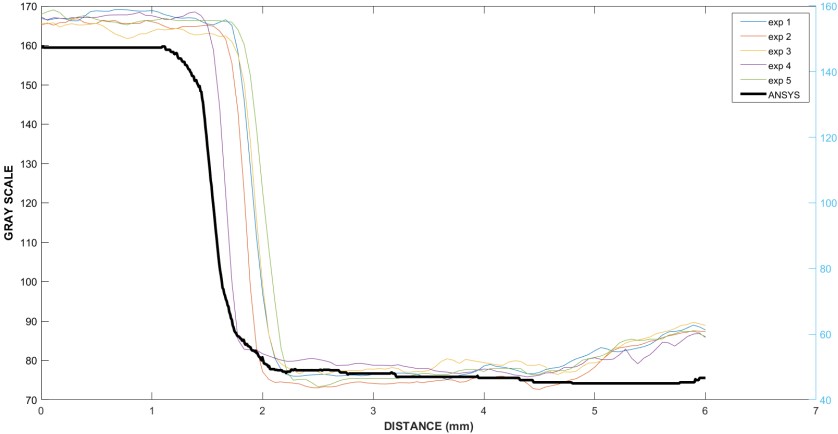

**Figure 15.** Comparative graph of ANSYS simulation and photo of the mixing process.

## 4. Conclusions

Our study demonstrates the remarkable precision of stereolithography (SLA) technology coupled with effective post-processing cleaning, allowing for the fabrication of microchannels as small as 200 μm in width and height. This breakthrough offers the exciting possibility for the creation of intricate microfluidic structures essential for capillary-driven devices. The device has a smaller channel of 502.74 μm with a standard deviation of 3.235 μm (designed as 500 μm). The mix tank has a width of 862.91 μm with a standard deviation of 12.82 μm (designed as 850 μm). The height of the printed channel has a less than 1% error according to the experimental measurements.

The utilization of capillary-driven systems proved highly effective in achieving the desired mixing quality. Notably, the integration of computational simulations played a pivotal role in predicting the future behavior and optimizing the design of the capillary device, ensuring efficient mixing performance. The experimental findings reveal an average velocity of 13.45 mm/s in channels with dimensions of 500 μm width × 400 μm height, while a reduced speed of 3.033 mm/s is observed in the mix tank featuring channels with dimensions of 850 μm width × 800 μm height. During the mixing process, there is a

notable increase in velocity, reaching an average of 15.09 mm/s within the mix tank. This enhancement in velocity signifies improved efficiency in the mixing process.

The successful mixing results obtained in this study have broad applications across diverse industries, including biomedicine, food processing, and more. These findings highlight the feasibility of manufacturing efficient mixing systems through SLA additive manufacturing, offering potential solutions to address specific industry challenges and demands. The manufacturing time for the device averages 90 min, with a maximum error of 2.6 % observed in the width of the channel-designed device and less than 1% in its height.

A noteworthy achievement of this research is the rapidity of fluid mixing within a relatively short microchannel length of 6 mm, with a duration ranging between 3 and 8 s. This efficiency is particularly notable when compared to the existing literature, underlining the effectiveness of our capillary-driven approach in achieving swift and effective mixing. The mixing index obtained through both the simulation and experiments was 0.55, referring to the fluid within the main channel.

**Author Contributions:** V.H.C.-M. executed the conceptualization, investigation, data analysis, manuscript preparation and wrote the original draft; J.C.-T. supervised, reviewed and edited the manuscript. All authors have read and agreed to the published version of the manuscript.

**Funding:** This work is funded by Universidad Politécnica Salesiana through Research Project (035-01-2024-01-30). Additionally, it received support from the Ministerio de Ciencia e Innovación (PID2020-114070RB-I00), the Agencia Estatal de Investigación (CPP2021-009021), and AGUAR (2021PROD00064).

**Institutional Review Board Statement:** Not applicable

**Informed Consent Statement:** Not applicable

**Data Availability Statement:** All data generated or analyzed during this study are included in this published article or available from the corresponding author upon reasonable request.

**Acknowledgments:** We gratefully acknowledge ESPE (Universidad de las Fuerzas Armadas) for providing the facilities and resources for the Scanning Electron Microscopy (SEM) microscopy, particularly the Nanomaterials Characterization Laboratory Center of Nanoscience and Nanotechnology (CENCINAT). Special thanks to Alexis Debut. We also appreciate the technical assistance Ing. Karla Vizuete for their guidance and support during the process.

**Conflicts of Interest:** The authors declare no conflicts of interest.

## Abbreviations

The following abbreviations are used in this manuscript:

| | |
|---|---|
| SLA | Stereolithography |
| FDM | Fused Deposition Modeling |
| UV | Ultraviolet |
| PP | Polypropylene |
| IGES | Initial Graphics Exchange Specification |
| SEM | Scanning Electron Microscopy |

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
