# Peer review of "Capillary-Driven Microdevice Mixer Using Additive Manufacturing (SLA Technology)"

_applsci, doi:10.3390/app14104293_

Round 1
Reviewer 1 Report
Comments and Suggestions for Authors
The current manuscript characterizes the performance of additively manufactured capillary-driven microdevice mixer fabricated by SLA technique. The manuscript is well-written and the materials and methods are clearly presented. However, there are some issues to be considered as follows:
- The "SLA" abbreviation should be included in the title as a complete expression.
- It is recommended to define the abbreviation through the text instead of displaying the abbreviations list by the end of the manuscript.
how many samples were fabricated and tested? what about the standard deviation in the measurements in case of testing more than one sample?
The conclusion should include the main results, contribution, and novelty of the current study using a bullet points style.
Comments on the Quality of English LanguageModerate change is required.
Author Response
Dear Reviewer,
I appreciate the time and feedback you provided for my research article. With this document, I am submitting the responses to your observations. Most of your comments and suggestions have been considered and incorporated into the final draft. I am looking forward to receiving a favorable response.
Best Regards

Reviewer 2 Report
Comments and Suggestions for Authors
This study proposes a passive device suitable for mobile chip laboratories to eliminate the need for external power supply. Using SLA technology to prepare complex geometric shapes to explore common laminar flow behavior in micro devices. After review, the following problems exist:
1. The VIDEO AND IMAGE PROCESS section, the specific parameters and equipment models for image acquisition need to be further explained. Image processing methods require further accurate and detailed descriptions, including steps, etc. You can refer to the following articles:Journal of Materials Science & Technology. 2024. 177: 44-58; IEEE Signal Processing Magazine, 2023, 40(4): 61-71.
2. How is the fluid velocity obtained in this article? How to determine its readiness?
3. The comparison of the simulation results with the experimental results in Fig. 14. needs to be further elaborated in the manuscript.
4 Two sections 3.3.4 appear in the manuscript.
Comments on the Quality of English LanguageModerate editing of English language required
Author Response

(The authors gave the same response as above.)

Reviewer 3 Report
Comments and Suggestions for Authors
The manuscript deals with the stereolithography (SLA) of a novel microfluidic mixer designed to blending two fluids with high efficiency. Authors have systematically studied the roles of the structure of Micro-mixers in performance, and simulated the mixing process. Authors have concluded that the process time is remarkably swift at 3 seconds, with a working flow rates range from 1 µL/s to 37 µL/s. However it would be great if the presentation of the Figures in experimental section can be refined.
Overall the study is rigorous and manuscript is well written. It can be published with minor revision.
Author Response

(The authors gave the same response as above.)
